# Skill-based Safe Reinforcement Learning with Risk Planning

## Abstract

Safe Reinforcement Learning (Safe RL) aims to ensure safety when an RL agent conducts learning by interacting with real-world environments where improper actions can induce high costs or lead to severe consequences. In this paper, we propose a novel Safe Skill Planning (SSkP) approach to enhance effective safe RL by exploiting auxiliary offline demonstration data. SSkP involves a two-stage process. First, we employ PU learning to learn a skill risk predictor from the offline demonstration data. Then, based on the learned skill risk predictor, we develop a novel risk planning process to enhance online safe RL and learn a risk-averse safe policy efficiently through interactions with the online RL environment, while simultaneously adapting the skill risk predictor to the environment. We conduct experiments in several benchmark robotic simulation environments. The experimental results demonstrate that the proposed approach consistently outperforms previous state-of-the-art safe RL methods.

## 1 Introduction

Reinforcement Learning (RL) empowers the development of intelligent agents and the training of decision systems, making it highly suitable for real-world applications. As RL continues to find broader use in real-world scenarios, concerns regarding the safety of RL systems have become more noticeable. These safety concerns have been particularly highlighted in human-centric domains, such as autonomous driving (Wen et al., 2020), helicopter manipulation (Koppejan & Whiteson, 2011), and human-related robotic environments (Brunke et al., 2021), where significant risks can be associated with taking improper actions, leading to severe consequences.

Safe Reinforcement Learning (Safe RL) focuses on the development of RL systems while adhering to predefined safety constraints (Garcıa & Fernández, 2015) and reducing the associated risk. In Safe RL, in addition to optimizing a reward function (Sutton & Barto, 2018), an additional cost is often assigned to evaluate the safety of actions taken by the RL agent; the RL agent aims to maximize the reward signal while ensuring a low cost (Altman, 1999; Hans et al., 2008). Conventional Safe RL methods aim to maximize cumulative rewards through interactions with online environments (Achiam et al., 2017; Tessler et al., 2019; Thomas et al., 2021), which often incur nontrivial costs in the learning process. More recently, researchers have recognized the value of learning from offline data, a practice that avoids potential damage to online physical environments (Xu et al., 2022; Liu et al., 2023). Reinforcement Learning from Demonstration (LfD) seeks to accelerate RL training by initially pre-training the RL agent using an offline dataset of demonstrations, which has demonstrated effective performance for standard RL tasks (Argall et al., 2009; Brys et al., 2015). Recent research has started to exploit the potential of LfD in the context of Safe RL, aiming to incorporate the safety-related information from the demonstration data to improve the training of safe policies in online environments (Thananjeyan et al., 2021). Our research endeavors to further advance safe RL in this intriguing direction.

In this paper, we introduce a novel Safe Skill Planning (SSkP) approach to enhance effective safe online RL by exploiting the offline demonstration data. Skill learning is a commonly used technique for LfD, allowing the RL agent to learn high-level representations of action sequences from offline demonstrations (Pertsch et al., 2021). In SSkP, we first employ a skill model to capture the high level behaviour patterns in the offline demonstrations as latent skills, and learn a skill risk predictor through Positive-Unlabeled (PU) learning on the demonstration data. The skill risk predictor

estimates the level of risk associated with executing a skill-based action sequence in a given state. Subsequently, we use the skill risk predictor to evaluate the safety of an RL agent's exploration behaviors (skills), and develop a novel risk planning process to enhance safe exploration and facilitate the efficient learning of a safe policy through interactions with online RL environments, while adapting the skill risk predictor to these online environments in real-time. We conduct experiments in various robotic simulation environments (Thomas et al., 2021) built on Mujoco (Todorov et al., 2012). The experimental results demonstrate that our proposed approach produces superior performance over several state-of-the-art safe RL methods, such as Recovery RL (Thananjeyan et al., 2021), CPQ (Xu et al., 2022) and SMBPO (Thomas et al., 2021). Our main contributions can be summarized as follows:

- We propose an innovative skill risk prediction methodology for extracting safe decision evaluation information from offline demonstration data and facilitating safe RL in online environments with planning.

- We devise a simple but novel risk planning process aimed at generating safer skill decisions by leveraging skill risk prediction, thereby enhancing safe exploration and learning in online RL environments.

- The proposed method SSkP demonstrates state-of-the-art safe RL performance.

## 2 RELATED WORKS

**Safe RL**    Safe Reinforcement Learning (Safe RL) is the study of optimizing decision-making for RL systems while ensuring compliance with safety constraints. It aims to strike a balance between exploration for learning and the avoidance of actions that could result in harmful or undesirable outcomes (García & Fernández, 2015). Altman (1999) first introduced the formulation of Constrained Markov Decision Processes (CMDPs) to frame the Safe RL problem. Subsequent research in (Hans et al., 2008) introduced strict constraints that prohibit safety violations within a single exploration trajectory. Thomas et al. (2021) developed a Safe Model-Based Policy Optimization (SMBPO) method, aiming to learn a precise transition model that prevents unsafe states during exploration by penalizing unsafe trajectories. Recent studies have highlighted the significance of incorporating offline data into Safe RL. Xu et al. (2022) introduced Constrained Penalized Q-learning (CPQ), which employs a cost critic to learn constraint values during exploration. They further penalize the Bellman operator in policy training to stop the update of the policy for potentially unsafe states. In another endeavor, Thananjeyan et al. (2020) proposed the Safety Augmented Value Estimation from Demonstrations (SAVED) approach, facilitating the learning of a safety density model from offline demonstration data. They utilize the cross-entropy method (Botev et al., 2013) for planning safe exploration, balancing task-driven exploration with cost-driven constrained exploration. Their more recent work introduced a Recovery RL approach (Thananjeyan et al., 2021), learning a recovery policy from offline demonstration data. This method ensures a recovery policy's safety by leveraging demonstration data, while also learning a recovery set to evaluate state safety. During online training, a task policy is learned when states are deemed safe, switching to the recovery policy when the RL agent encounters potentially unsafe situations.

**Skill-based RL**    Reinforcement Learning from Demonstration (LfD), also known as Imitation Learning, focuses on enhancing online RL training by leveraging an expert demonstration dataset (Argall et al., 2009; Brys et al., 2015). Thrun & Schwartz (1994) introduced skill learning to LfD, enabling RL agents to learn reusable high-level skills from action sequences within offline demonstration data. In more recent research, Pertsch et al. (2021) presented the SPiRL framework, which leverages deep latent models to learn skill representations. The policy is trained using the skill model in conjunction with a variant of Soft Actor-Critic (SAC) (Haarnoja et al., 2018) to accelerate RL in downstream tasks. Furthermore, recent work has demonstrated the integration of skill learning into *offline* safe RL (Slack et al., 2022), which learns a safety variable posterior from offline demonstration data and subsequently enhances online safe policy training.

**Positive-Unlabeled Learning**    In contrast to traditional supervised learning that relies on labeled positive and negative examples, Positive-Unlabeled (PU) learning addresses scenarios where data cannot be strictly categorized as positive or negative. Notably, Du Plessis et al. (2014; 2015)'s

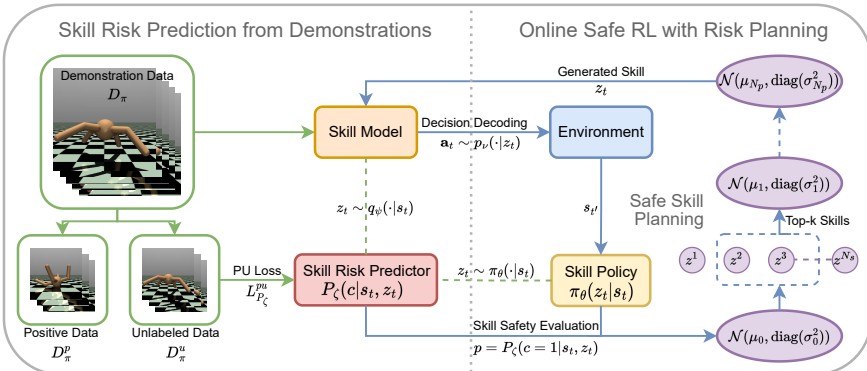

Figure 1: The framework of the proposed method, SSkP, which learns a skill risk predictor from the offline demonstration data and then deploys it to enhance online safe RL through risk planning. During the skill risk predictor learning stage, SSkP assembles PU data and trains a decision risk predictor $P_\zeta(c|s_t, z_t)$ based on a skill model, which produces skill prior $q_\psi(\cdot|s_t)$ and skill decoder $p_\nu(\mathbf{a}_t|z_t)$. In the online safe policy learning stage, a risk planning process is deployed to generate and choose safer skill decisions based on the skill risk predictor $P_\zeta(c|s_t, z_t)$. The generated skill $z_t$ is decoded by the skill decoder $p_\nu(\mathbf{a}_t|z_t)$ into an action sequence $\mathbf{a}_t$ to interact with the online environment. Rewards are collected from online interactions to learn the safe skill policy $\pi_\theta(z_t|s_t)$.

previous work introduced an unbiased estimation of the true negative loss, making PU learning feasible. Jain et al. (2016) and Christoffel et al. (2016) extended this research by enhancing the accuracy of practical PU classifier training through positive class prior estimation. Kiryo et al. (2017) proposed a large-scale PU learning approach that addresses overfitting by introducing non-negative constraints and a relaxed slack variable. In recent developments, Xu & Denil (2021) applied PU learning to Generative Adversarial Imitation Learning (GAIL) (Ho & Ermon, 2016) in RL, which learns an optimized reward function from the expert demonstration dataset to improve RL performance in offline training.

## 3 PROBLEM SETTING

The safe RL problem is typically framed as a Constrained Markov Decision Process (CMDP) (Altman, 1999), denoted as $M = (\mathcal{S}, \mathcal{A}, \mathcal{T}, \mathcal{R}, \mathcal{C}, \gamma)$, where $\mathcal{S}$ represents the state space, $\mathcal{A}$ is the action space, $\mathcal{T} : \mathcal{S} \times \mathcal{A} \to \mathcal{S}$ defines the transition dynamics, $\mathcal{R} : \mathcal{S} \times \mathcal{A} \to \mathbb{R}$ is the reward function, and $\gamma \in (0, 1)$ is the discount factor. The additional cost function $\mathcal{C} : \mathcal{S} \times \mathcal{A} \to \mathbb{R}$ is introduced to account for safety violations during RL exploration. Hence an exploration trajectory within CMDP can be expressed as $\tau = (s_0, a_0, r_0, c_0, \ldots, s_t, a_t, r_t, c_t, \ldots, s_{|\tau|+1})$. We adopt the strict setting that the safe RL agent will terminate a trajectory when encountering safety violation and inducing a nonzero cost ($c_t > 0$) (Hans et al., 2008). The goal of safe RL is to efficiently learn a good policy $\pi$ that maximizes expected discounted cumulative reward while incurring minimal costs.

To facilitate safe RL in online environments, we presume the availability of a small demonstration dataset, denoted as $\mathcal{D}_d$, which provides prior information regarding safety violations during exploration: $\mathcal{D}_d = \{\ldots, (\cdots, s_t, a_t, c_t, \cdots), \ldots\}$. The demonstration data can be gathered by either human experts or a trained safe RL agent (Thananjeyan et al., 2021). A method that can effectively exploit such demonstration data is expected to accelerate safe RL in online environments with smaller costs.

## 4 METHOD

The main framework of the proposed Safe Skill Planning (SSkP) approach is presented in Figure 1, which has two stages: skill-risk predictor learning and safe RL with risk planning. Towards the goal of facilitating efficient safe RL, SSkP first exploits the prior demonstration data to extract reusable high-level skills and learn a skill risk predictor through PU learning. Then by devising a

risk planning process based on the skill risk predictor, the online RL agent is guided to pursue risk-averse explorations and efficiently learn a skill policy in online environments that can maximize the expected reward with minimal costs. We further elaborate these two stages in the following subsections.

## 4.1 SKILL RISK PREDICTION FROM DEMONSTRATIONS

Conventional safe RL methods entail the learning of a safe policy through direct interaction with the online environment, which often incur considerable costs in the exploration based learning process. Learning from demonstration (LfD) offers a means to accelerate the online RL process and reduce the cost by pre-training on an offline demonstration dataset. This pre-training phase is more efficient in terms of time and cost compared to the resource-intensive online environment. Skill-based learning stands as a prominent approach in LfD (Pertsch et al., 2021). It learns reusable skills as generalizable high level representations of action sequences from offline demonstrations, which can be used to guide the RL agent to explore in a safe manner for downstream online tasks. Inspired by the principles of LfD, we aim to extract skill-based safety-related insights from the demonstration dataset $\mathcal{D}_d$, which can be utilized to assess the safety of reinforcement decisions and enhance the ensuing online safe RL. In particular, we propose to learn a skill risk predictor $P_\zeta(c|s_t, \boldsymbol{z}_t)$ from the demonstration data that can evaluate the safety of a skill-based decision, $\boldsymbol{z}_t$, on a given state $s_t$.

To support skill-based learning, we first adopt the deep skill model from a previous work (Pertsch et al., 2021) to learn skills as latent representations of observed action sequences. This skill model consists of three key components: a skill encoder network $q_\mu(\boldsymbol{z}_t|\mathbf{a}_t)$, responsible for encoding an action sequence $\mathbf{a}_t = \{a_t, ..., a_{t+H-1}\}$ with length $H$ into a high-level skill $\boldsymbol{z}_t$; a skill decoder network $p_\nu(\mathbf{a}_t|\boldsymbol{z}_t)$, which decodes the skill $\boldsymbol{z}_t$ back into the action sequence $\mathbf{a}_t$; and a skill prior network $q_\psi(\boldsymbol{z}_t|s_t)$, which generates the skill decision for a given state $s_t$. After being trained on the demonstration data $\mathcal{D}_d$, the components of the skill model can be deployed to facilitate subsequent learning processes.

### 4.1.1 LEARNING SKILL RISK PREDICTOR VIA PU LEARNING

The demonstration data provides valuable insights for safe exploration of the environment. However, estimating risk predictors for skill-based behaviors in the context of safe exploration poses a persistent challenge due to two primary reasons. First, the demonstration data, whether collected by a human expert or a fully trained safe RL agent, often contain very limited actual examples of safety violations, due to the finite trajectory lengths and limited skill horizons. Second, while a decision made in a given state may not result in immediate safety violations, it could lead to a close proximity to safety violations. Treating such decisions as strictly safe examples can be problematic. To tackle these issues, we propose the utilization of Positive-Unlabeled (PU) learning, a technique that can bypass the strict differentiation of safe decisions from unsafe ones and alleviate the scarcity of unsafe examples.

Specifically, we collect the positive and unlabeled decision examples for PU learning as follows. At a timestep $t$, if the current trajectory $\tau$ actually encounters a safety violation within the next $H$ steps when the RL agent is projected to select skill $\boldsymbol{z}_t$ at state $s_t$, then we collect such state-skill pair $(s_t, \boldsymbol{z}_t)$ as *positive* unsafe examples. Conversely, all other state-skill decision pairs that do not lead to immediate risks are collected as *unlabeled* examples. For states near the termination of trajectories, the corresponding action sequences have lengths that are insufficient (less than the horizon $H$) to encode skills. We hence utilize the skill prior network $q_\psi(\boldsymbol{z}_t|s_t)$ from the skill model to produce the skill decision $\boldsymbol{z}_t$ for each given state $s_t$ in the demonstration data, instead of using the encoder.

Let $D^p = (s_t^p, \boldsymbol{z}_t^p)$ represent the set of positive examples of state-skill decision pairs, and $D^u = (s_t^u, \boldsymbol{z}_t^u)$ represent the unlabeled set. We learn the skill risk predictor $P_\zeta(c|s_t, \boldsymbol{z}_t)$ as a binary classifier parameterized with $\zeta$, measuring the probability of selecting skill $\boldsymbol{z}_t$ at state $s_t$ leading to a safety violation with risk $c > 0$. We compute the true positive loss on the PU training data as the negative mean log-likelihood of the positive examples in $D^p$:

$$L_{P_\zeta}^1(D^p) = -\mathbb{E}_{(s,\boldsymbol{z})\sim D^p}[\log(P_\zeta(c = 1|s, \boldsymbol{z}))], \tag{1}$$

while the difficulty lies in computing the true negative loss without confirmed negative examples. To bypass this problem, unbiased estimation of the true negative loss using PU data has been developed

---

**Algorithm 1** Risk Planning

---

**Initialize:** $(\boldsymbol{\mu}_0, \boldsymbol{\sigma}_0^2) \leftarrow q_\psi(\cdot|s_t)$
**Procedure:**
1: **for** $i = 1, 2, ..., N_p$ **do**
2:     Sample skills $\{\boldsymbol{z}^j\}_{j=1}^{N_s}$ from $\mathcal{N}(\boldsymbol{\mu}_{i-1}, \mathrm{diag}(\boldsymbol{\sigma}_{i-1}^2))$
3:     Calculate $p_j = P_\zeta(c = 1|s_t, \boldsymbol{z}^j)$ for $N_s$ skills
4:     Compute $(\boldsymbol{\mu}_i, \boldsymbol{\sigma}_i^2)$ using the selected top-k skills with lowest risk predictions in $\{p_j\}_{j=1}^{N_s}$
5: **end for**
6: Sample skill $\boldsymbol{z}_t$ from $\mathcal{N}(\boldsymbol{\mu}_{N_p}, \mathrm{diag}(\boldsymbol{\sigma}_{N_p}^2))$

---

in the literature (Du Plessis et al., 2015; 2014):

$$L_{P_\zeta}^0(D^u \cup D^p) = L_{P_\zeta}^0(D^u) - \lambda L_{P_\zeta}^0(D^p) \tag{2}$$

where $\lambda$ represents the positive class prior, which can be estimated using positive and unlabeled data (Jain et al., 2016; Christoffel et al., 2016); $L_{P_\zeta}^0(D)$ denotes the negative expectation of the log-likelihood of the given data $D$ being negative, such that:

$$L_{P_\zeta}^0(D) = -\mathbb{E}_{(s,\boldsymbol{z})\sim D}[\log(1 - P_\zeta(c = 1|s, \boldsymbol{z}))]. \tag{3}$$

To further improve the estimation of the true negative loss, in the recent PU learning literature, Kiryo et al. (2017) introduce an additional constraint to the estimation of $L_{P_\zeta}^0(D^u \cup D^p)$, ensuring that the loss remains non-negative: $L_{P_\zeta}^0(D^u) - \lambda L_{P_\zeta}^0(D^p) \geq 0$. To provide tolerance and reduce the risk of overfitting, a non-negative slack variable $\xi \geq 0$ is also introduced to relax the constraint, which leads to the following PU loss we adopted for training our skill risk predictor:

$$L_{P_\zeta}^{pu}(D^p, D^u) = \lambda L_{P_\zeta}^1(D^p) + \max(-\xi, L_{P_\zeta}^0(D^u) - \lambda L_{P_\zeta}^0(D^p)) \tag{4}$$

By minimizing this PU loss on the demonstration data, we obtain a pre-trained skill risk predictor $P_\zeta(c|s_t, \boldsymbol{z}_t)$, which will be deployed in the online RL stage to screen the skill decisions and accelerate safe policy learning.

## 4.2 ONLINE SAFE RL WITH RISK PLANNING

In the online safe policy learning stage, our objective is to facilitate the learning of a safe policy by leveraging the safe skill knowledge learned from the offline demonstration data, encoded by the skill prior network $q_\psi(\cdot|s_t)$, the decoder network $p_\nu(\cdot|\boldsymbol{z}_t)$, and, in particular, the skill risk predictor $P_\zeta(c|s_t, \boldsymbol{z}_t)$.

### 4.2.1 RISK PLANNING

The pre-trained skill risk predictor $P_\zeta(c|s_t, \boldsymbol{z}_t)$ encodes safe decision evaluation information extracted from the demonstration data, providing an essential capacity for pre-assessing the safety of potential skill-based decisions before executing them in online environments. Specifically, $P_\zeta(c = 1|s_t, \boldsymbol{z}_t)$ can quantify the likelihood that the RL agent will encounter safety violation by following the action sequence encoded by skill $\boldsymbol{z}_t$ at state $s_t$. We have, therefore, developed a simple heuristic risk planning process that leverages the skill risk predictor to choose safer skill decisions to follow. This process is expected to reduce the potential for encountering safety violations and enhance the safety of online RL learning.

Specifically, we evaluate and choose skill-based decisions at a given state $s_t$ from an iteratively self-enhanced Gaussian distribution $\mathcal{N}(\boldsymbol{\mu}, \mathrm{diag}(\boldsymbol{\sigma}^2))$ that has a diagonal covariance matrix. At the start, we sample $N_s$ skills from the current safe policy function $\pi_\theta(\cdot|s_t)$ at state $s_t$ such that $\{\boldsymbol{z}^j \sim \pi_\theta(\cdot|s_t), j = 1 \cdots N_s\}$, and use these skill vectors to calculate the mean and covariance of an initial Gaussian distribution $\mathcal{N}(\boldsymbol{\mu}_0, \mathrm{diag}(\boldsymbol{\sigma}_0^2))$. Then in each $i$-th iteration, we sample $N_s$ skills $\mathcal{Z} = \{\boldsymbol{z}^j\}_{j=1}^{N_s}$ from the current Gaussian distribution $\mathcal{N}(\boldsymbol{\mu}_{i-1}, \mathrm{diag}(\boldsymbol{\sigma}_{i-1}^2))$ and evaluate their safety using the skill risk predictor $p_j = P_\zeta(c|s_t, \boldsymbol{z}^j)$. We choose the top-k safe skills $\mathcal{Z}_k$ with the lowest

---

**Algorithm 2** Online Safe Policy Learning

---

**Input:** skill prior $q_\psi(\cdot|s)$, decoder $p_\nu(\cdot|\boldsymbol{z})$,  skill risk predictor $P_\zeta(c|s,\boldsymbol{z})$, $D^p$ and $D^u$
**Initialize:** data buffer $D$, skill policy network $\pi_\theta(\boldsymbol{z}|s)$
**Procedure:**
1: **for** each episode **do**
2:    Randomly start from a state $s_0$,   set $t = 0$
3:    **for** every $H$ environment steps **do**
4:       $\boldsymbol{z}_t \leftarrow \text{Risk\_Planning}(\pi_\theta(\cdot|s_t), P_\zeta(c|s_t, \boldsymbol{z}))$
5:       Sample $\mathbf{a}_t = a_{t:t+H-1}$ from decoder $p_\nu(\cdot|\boldsymbol{z}_t)$
6:       Execute $\mathbf{a}_t$: stop current trajectory $\tau$ when $c>0$
7:       Collect reward $\tilde{r}_t$ and get next state $s_{t'}$
8:       Add $\{s_t, \boldsymbol{z}_t, \tilde{r}_t, s_{t'}\}$ to $D$ with $t'= t+\min(H, |\tau|)$
9:       Collect decision pairs $\mathcal{P}$ as in Eq.(7)
10:      **If** $c > 0$ **then:** Add $\mathcal{P}$ to $D^p$    **else:** Add $\mathcal{P}$ to $D^u$    **end if**
11:      **If** $c > 0$ or reached max episode-steps    **then**   break out    **end if**
12:      $t = t'$
13:   **end for**
14:   Update predictor $P_\zeta(c|s, \boldsymbol{z})$ by minimizing Eq. (4)
15:   Update policy network $\theta$ following the skill-based SAC method on $D$.
16: **end for**

---

predicted risk probabilities from $\mathcal{Z}$ to update the Gaussian distribution for the next iteration:

$$\boldsymbol{\mu}_i = \frac{1}{k} \sum_{\boldsymbol{z} \in \mathcal{Z}_k} \boldsymbol{z}, \tag{5}$$

$$\boldsymbol{\sigma}_i^2 = \frac{1}{k} \sum_{\boldsymbol{z} \in \mathcal{Z}_k} \text{diag}\left((\boldsymbol{z} - \boldsymbol{\mu}_i)(\boldsymbol{z} - \boldsymbol{\mu}_i)^\top\right) \tag{6}$$

After a total number of $N_p$ iterations, an optimized skill decision $\boldsymbol{z}_t$ with low predicted risk is sampled from the final refined distribution $\mathcal{N}(\boldsymbol{\mu}_{N_p}, \text{diag}(\boldsymbol{\sigma}_{N_p}^2))$. The procedure of this planning process is also summarized in Algorithm 1. This risk planning procedure is essentially a cross-entropy method (CEM) (Botev et al., 2013; Rubinstein, 1997), specifically employed in this context as a zeroth-order solver to tackle the non-convex optimization problem (Amos & Yarats, 2020) of $\arg\min_{\boldsymbol{z}} P_\zeta(c = 1|s_t, \boldsymbol{z})$, facilitating effective selection of safe skills based on the skill risk predictor. By gradually adjusting the Gaussian distribution towards safer decision skill regions, we expect to reliably identify a safe skill to deploy after a sufficient number of iterations.

### 4.2.2 ONLINE SAFE POLICY LEARNING

By utilizing the pre-learned skill knowledge and the proposed risk planning process, we aim to efficiently learn a skill-based safe policy network $\pi_\theta(\boldsymbol{z}|s)$ through iterative interactions with an online RL environment, which maximizes the expected discounted reward while minimizing the costs incurred by safety violations. Specifically, at the current state $s_t$, we first select an optimized skill, $\boldsymbol{z}_t$, using the risk planning process. This skill, $\boldsymbol{z}_t$, is then decoded into an action sequence, $\mathbf{a}_t = a_{t:t+H-1}$, using the skill decoder $p_\nu(\cdot|\boldsymbol{z}_t)$. The RL agent interacts with the online environment to reach next state $s_{t'}$ by taking this sequence of actions, adhering to the behavior patterns of the pre-learned skills. Such skill-based planning and decision making are more efficient to carry on as well than single actions. During the interaction process, the RL agent collects cumulative reward signals $\tilde{r}_t = \sum_t^{t'-1} r_t$ from the environment and monitors the cost signal $c$, which will become positive ($c > 0$) when encountering safety violation. The trajectory will be terminated with safety violation without executing the whole sequence of actions. Without safety violation, the next state reached from $s_t$ will be $s_{t'} = s_{t+H}$. The skill-based transition data, $D = \{(s_t, \boldsymbol{z}_t, \tilde{r}_t, s_{t'})\}$, are collected from the online interactions to train the safe skill policy $\pi_\theta(\cdot)$. Meanwhile the state-skill decision pairs are collected as PU examples in a similar way as on the demonstrations, such that

$$\mathcal{P} = \{(s_t, \boldsymbol{z}_t), (s_i, \boldsymbol{z} \sim q_\psi(\cdot|s_i))|i \in \{t+1 : t'-1\}\}. \tag{7}$$

These are then integrated with the existing PU data to continuously adapt the skill risk predictor to the online environment in *real-time*, enhancing and accelerating the online safe RL policy learning. The full procedure of the proposed online safe RL learning is presented in Algorithm 2.

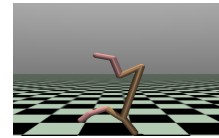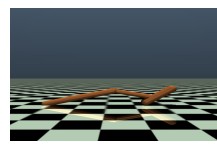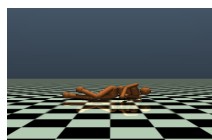

Figure 2: The figures present instances of failure in each environment where safety constraints are violated. From left to right: *Ant, Cheetah, Hopper, Humanoid*.

In this work, we deploy a skill-based Soft Actor-Critic (SAC) algorithm (Haarnoja et al., 2018) to learn the skill policy network $\pi_\theta(\cdot)$ on the collected data $D$, which enforces behavior cloning by replacing the entropy regularizer in the optimization objective of SAC with a KL-divergence regularizer, $KL(\pi_\theta(\cdot|s), q_\psi(\cdot|s))$, between the skill policy network $\pi_\theta(\cdot|s)$ and the pre-trained prior network $q_\psi(\cdot|s)$ (Pertsch et al., 2021).

# 5 EXPERIMENT

## 5.1 EXPERIMENTAL SETTINGS

**RL Environments**   We conducted experiments with four benchmark robotic simulation environments, namely, *Ant, Cheetah, Hopper,* and *Humanoid*, utilizing a customized variant of the MuJoCo physics simulator (Todorov et al., 2012) as introduced in (Thomas et al., 2021). In these environments, the RL agent halts upon encountering a safety violation. In the *Ant* and *Hopper* environments, a safety violation occurs when the robot topples over. In the *Cheetah* environment, a violation takes place when the robot's head hits the ground. In the *Humanoid* environment, the human-like robot violates the safety constraint when its head falls to the ground. Figure 2 presents some instances of failure in these environments. The RL agent is trained to maximize cumulative rewards while adhering to the safety constraint.

**Comparison Methods**   We compare our proposed SSkP approach with three state-of-the-art safe RL methods: CPQ (Xu et al., 2022), SMBPO (Thomas et al., 2021), and Recovery RL (Thananjeyan et al., 2021). CPQ is a constraints penalized Q-learning method. It learns from offline demonstration data, and penalizes the Bellman operator during policy training when encountering unsafe states. SMBPO is a model-based method that relies on an ensemble of Gaussian dynamics-based transition models. It penalizes trajectories that lead to unsafe conditions and avoids unsafe states under specific assumptions. Recovery RL first learns a recovery policy from the offline demonstration data with the objective of minimizing safety violations. During online training, the agent takes actions to maximize the reward signal in safe situations and falls back on the recovery policy to reduce safety violations if necessary.

**Implementation Details**   A fixed horizon length $H = 10$ for skill action sequences is used in the experiments. The dimension of the skill vectors is set as 10. The PU risk predictor, skill decoder, and policy network employ standard MLP architectures, while the skill prior incorporates an MLP with a Gaussian output layer. The skill encoder utilizes an LSTM with linear output. Following prior work on PU learning (Xu & Denil, 2021), the slack variable $\xi$ is set to 0. For risk planning, we used $N_s = 512$, $k = 64$, and $N_p = 6$. For comparison, we used the official implementations of Recovery RL (Thananjeyan et al., 2021) and SMBPO (Thomas et al., 2021). The implementation of CPQ is adapted from the OSRL repository (Liu et al., 2023). In the case of Recovery RL, both offline and online components are enabled. As for CPQ (Xu et al., 2022), the agent is pre-trained on the same offline dataset we collected and then trained in the same manner in online environments. All results are collected over a total of $10^6$ online timesteps.

## 5.2 EXPERIMENTAL RESULTS

The comparison results for our proposed SSkP method and the other three safe RL methods in four robotic simulation environments are presented in Figure 3. We used a similar evaluation strategy as the one in (Thomas et al., 2021). The results for all the methods are collected over the same total of $10^6$ online timesteps. As the goal is to maximize the expected reward while minimizing the

Table 1: Comparison results in terms of the ratio between Per-timestep Reward (PtR) and #Violations (PtR/#V ($\times 10^3$)). This metric reflects the cost-sensitive sample efficiency of online safe RL.

|  | Ant | Cheetah | Hopper | Humanoid |
|---|---|---|---|---|
| SSkP | 23.54 | **173.72** | **8.86** | **0.72** |
| Recovery RL | 13.12 | 146.30 | 7.10 | 0.71 |
| CPQ | 11.80 | 92.25 | 6.21 | 0.38 |
| SMBPO | **28.68** | 147.00 | 5.74 | 0.69 |

Figure 3: This figure presents the performance of each comparison method in terms of the average episode reward vs. the total number of safety violations encountered during online training within a fixed total number of timesteps on all four environments: *Ant*, *Cheetah*, *Hopper*, and *Humanoid*. The results represent the averages over three runs, with the shadow indicating the standard deviations.

safety violation costs, we present the performance of each method in terms of its average episode reward versus the total number of safety violations encountered. Specifically, the x-axis depicts the cumulative safety violations encountered by the RL agent throughout the entire online training process, while the y-axis reflects the average episode rewards with the increasing of numbers of violations. These plots effectively illustrate the trade-off between reward maximization and risk (safety violation) minimization. A higher average episode reward with the same number of safety violations indicates better performance in policy learning with the same cost.

We can see that across all four environments, CPQ exhibits an initial advantage with a higher starting point and eventually halts with a very low average episode reward. This demonstrates that CPQ *failed to learn* a good policy function within the total $10^6$ online timesteps. Although it only encountered a lower total number of violations, the inability to effectively perform RL failed the ultimate goal. This can be attributed to that CPQ pre-trains its policy on the offline demonstration dataset. In contrast, both our proposed SSkP and Recovery RL do not rely on policy learning from offline demonstrations. SSkP learns the skill model and the skill risk predictor from the offline demonstration data and deploys them to support the online safe RL policy learning. SSkP outperforms Recovery RL in all four environments, producing much higher average rewards with lower numbers of safety violations. SSkP also largely outperforms SMBPO in a similar way in three out of the four environments, except for the *Ant* environment; in *Ant*, SMBPO demonstrates a similar inability as CPQ in terms of learning a good policy to maximize the expected reward. Overall, the proposed SSkP method produces the most effective performance in all the four environments, outperforming the other comparison methods. This validates the effectiveness of SSkP for advancing safe RL by exploiting offline demonstrations.

To provide a quantitative measure for the performance of an online safe RL agent throughout the entire online learning process, we further introduce a new metric to compute the ratio between the Per-timestep Reward (PtR) and the total number of safety Violations (#V), denoted as *PtR/#V*. PtR is calculated by dividing the cumulative episode reward across the entire online training duration by the total number of timesteps, which indicates the sample efficiency of the RL agent. Specifically, let $E$ represent the total number of episodes, $R_e$ denote the episode reward at episode $e$, $T$ denote the total number of timesteps. Then PtR is computed as $\sum_{e=1}^{E} R_e/T$. By further computing the ratio between PtR and the total number of safety violations, *PtR/#V* takes the safety into consideration and can be used as a *cost-sensitive sample efficiency metric* for safe RL, which can capture the tradeoff between the learning efficiency of the safe RL agent and the cost of encountering safety violations. The objective of safe RL is to maximize the reward while minimizing safety violation costs, naturally favoring a larger *PtR/#V* ratio value. We calculated the average *PtR/#V* values over three runs for all the comparison methods in all the four experimental environments, and reported the comparison results in Table 1, where the *PtR/#V* numbers are scaled at $10^3$ for clarity of presentation.

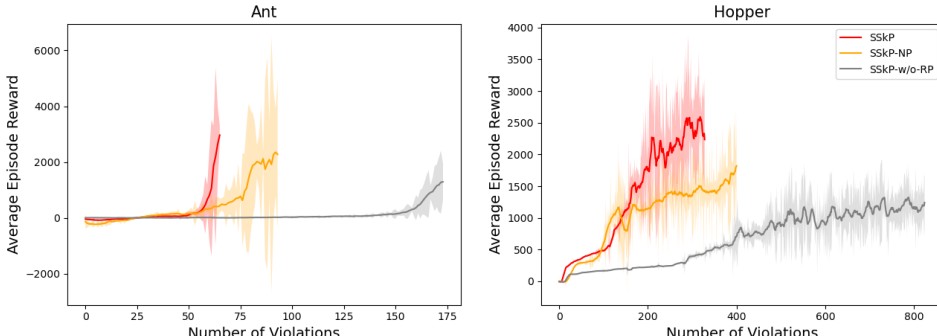

Figure 4: The ablation study results in two environments: *Ant* and *Hopper* by comparing three methods: SSkP—the proposed approach; SSkP-NP—the variant that replaces risk planning with a naive planning process; and SSkP-w/o-RP—the variant that drops risk predictor and risk planning from SSkP. Each plot displays the average reward vs. the total number of safety violations encountered during online training within a fixed total number of timesteps. The results are averages of three runs.

Notably, under the PtR/#V metric, our SSkP method outperforms all the other comparison methods in three out of the total four environments, except for the *Ant* environment, where SSkP produced the second-best result. The comparison method, CPQ, that has been shown to fail to learn in the figures, produces poor PtR/#V values in all the environments. Particularly in *Cheetah* and *Hopper*, SSkP produces notable performance gains over all the other methods. These results again validate the superior efficiency and efficacy of our SSkP for online safe RL.

## 5.3 ABLATION STUDY

The main contribution of the proposed SSkP approach lies in devising two novel components: the *risk planning* component and the *skill risk predictor*. We conducted an ablation study to investigate their impact on the performance of SSkP.

The risk planning component in SSkP iteratively improves the safety of skills by leveraging the skill risk predictor, aiming to generate and deploy the most effective safe skill decision. To investigate the extent to which the proposed risk planning process enhances safe policy learning performance, we introduced an alternative *naive planning* baseline as a comparison. Naive planning samples $N_s$ skills using the current safe policy $\pi_\theta(\cdot|s_t)$ at the given state $s_t$, evaluates them using the current skill risk predictor, and selects the best skill with the lowest predicted risk *in a single iteration*. We denote the variant of SSkP with naive planning instead of the proposed risk planning as *SSkP-NP*.

The SSkP-NP variant nevertheless still leverages the skill risk predictor. To further investigate the impact of the skill risk predictor, we introduced another variant, *SSkP-w/o-RP*, which drops the skill risk predictor learning and deployment from both the offline and online learning stages. Consequently, risk planning that depends on skill risk assessment is also disabled in the online RL stage, while the skill decisions are produced directly by the skill policy function.

We compared the proposed full approach SSkP with the two variants, SSkP-NP and SSkP-w/o-RP, in the *Ant* and *Hopper* environments, and the experimental results are presented in Figure 4. The curves in the figure reveal that our proposed SSkP with risk planning clearly outperforms the ablation variant SSkP-NP with naive planning in both environments. In the *Hopper* environment, SSkP-NP exhibits a very brief faster improvement during the early training stage but experiences a subsequent decline. Our proposed full approach SSkP produces a much better policy function that achieves substantially much higher average episode reward than SSkP-NP with smaller cost—the number of safety violations. This validates the contribution put forth by the proposed risk planning process. We also note that by eliminating the skill risk predictor and consequently the entire risk planning, the variant SSkP-w/o-RP, while still leveraging the offline demonstration data through the skill model, experiences a substantial performance decline compared to SSkP-NP. The results validate the significant contribution of the proposed skill risk prediction methodology, which is the foundation of the proposed safe RL method SSkP.

## 6 CONCLUSION

In this paper, we introduced a Safe Skill Planning (SSkP) method to address the challenge of online safe RL by effectively exploiting a prior demonstration dataset. First, we deployed a deep skill model to extract safe behavior patterns from the demonstrations and proposed a novel skill risk predictor for decision safety evaluation, which is trained through PU learning over the state-skill pairs. Second, by leveraging the risk predictor, we devised a new and simple risk planning process to iteratively identify reliable safe skill decisions in online RL environments and support online safe RL policy learning. We compared the proposed method with several state-of-the-art safe RL methods in four benchmark robotic simulation environments. The experimental results demonstrate that our method yields notable improvements over previous online safe RL approaches.

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

## A  ALTERNATIVE EVALUATION OF EXPERIMENTAL RESULTS

We have introduced an alternative evaluation of our experimental results in Section 5.2, simultaneously presenting sample efficiency curves and violation curves. This approach offers an intuitive understanding of the overall performance of our safe RL agent, illustrating performance and safety metrics across environmental steps. The results are illustrated in Figure 5. Notably, on the *Ant*,

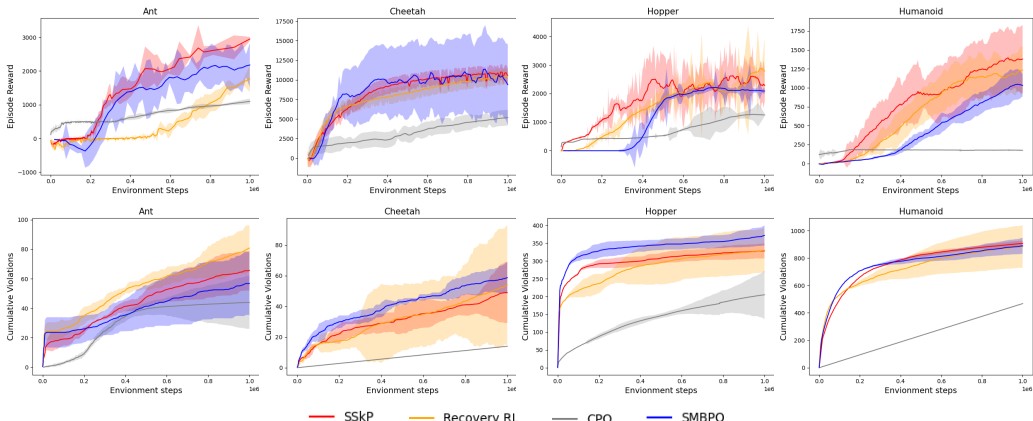

Figure 5: The alternative evaluation of the safe RL results presents the episode rewards and the cumulative number of violations separately along the environment steps. **Top:** Sample efficiency curves illustrating episode rewards v.s. the total number of environmental steps across four environments. **Bottom:** Violation curves illustrating the total number of violations v.s. the total number of environmental steps across four environments. The results are averages of three runs.

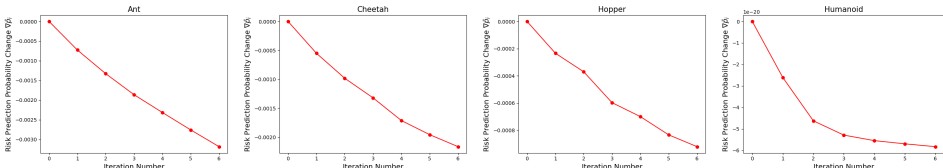

Figure 6: Risk prediction probability changes, $\nabla \bar{p}_i = \bar{p}_i - \bar{p}_0$, along the planning iteration number $i$ from the initial average risk prediction probability $\bar{p}_0$. The results are the averages computed with the risk planning procedure on 100 randomly sampled states $s_t$.

*Hopper*, and *Humanoid* environments, our SSkP demonstrates superior performance based on sample efficiency curves, while on the *Cheetah* environments, SSkP exhibits comparable performance to Recovery RL and SMBPO. These findings highlight SSkP's robust performance across environments, even in the absence of explicit safety constraints. Although CPQ displays the lowest cumulative violations compared to other methods, it fails to achieve acceptable episode rewards, indicating its incapacity to learn an effective policy while following safety constraints. For the *Cheetah*, *Hopper*, and *Humanoid* environments, as the number of environmental steps increases, SSkP exhibits comparable safety violations with the second-best comparison method (excluding CPQ), while outperforming comparison methods in terms of episode rewards.

## B  FURTHER STUDY OF RISK PLANNING PROCESS

The ablation study above validated the contribution of the proposed risk planning procedure towards our overall safe RL approach, SSkP. In this subsection, we further study the efficacy of the risk planning procedure in Algorithm 1 as a zeroth-order solver for the non-convex optimization problem of $\arg\min_{\boldsymbol{z}} P_{\zeta}(c = 1|s_t, \boldsymbol{z})$ by presenting the changes in the predicted risk probabilities of the sampled skills along the Gaussian distribution refinement iterations.

Specifically, in each experimental environment, given the trained risk predictor $P_{\zeta}(\cdot)$ and a sampled state $s_t$, we conduct risk planning with $N_p = 6$ refinement iterations. From each Gaussian distribution $\mathcal{N}(\boldsymbol{\mu}_i, \text{diag}(\boldsymbol{\sigma}_i^2))$, along the iterations $i \in \{0, 1, \cdots, N_p\}$, we sample $N_s$ skills $\{\boldsymbol{z}_i^j\}_{j=1}^{N_s}$ and calculate the average of their predicted risk probabilities, $\bar{p}_i = \frac{1}{N_s} \sum_j^{N_s} P_{\zeta}(c = 1|s_t, \boldsymbol{z}_i^j)$. To emphasize the effect of reducing risks of the sampled skills, we report the changes of the average risk probability from the initial iteration 0; i.e., we record $\nabla \bar{p}_i = \bar{p}_i - \bar{p}_0$ for each iteration $i$. We repeat this risk planning process over 100 randomly sampled states $\{s_t\}$, and report the average results in Figure 6 for all the four experimental environments. We can see with the increase of the risk planning

Table 2: The table presents experimental results of SSkP on Ant and Hopper environments in terms of the ratio between Per-timestep Reward (PtR) and #Violations (PtR/#V ($\times 10^3$)) at various proportions of the total offline data size. The results are averages over three runs.

| Proportion of Offline Data | 1.0 | 0.5 | 0.2 | 0.1 |
|---|---|---|---|---|
| Ant | 23.54 | 19.30 | 15.09 | 10.06 |
| Hopper | 8.86 | 7.96 | 6.17 | 4.94 |

iterations, $-\nabla \bar{p}_i$ becomes larger and hence $\bar{p}_i$ becomes smaller, indicating the sampled skills from each current Gaussian distribution are safer than previous iterations. Overall, the results validate that the risk planning process can effectively find safer skills $z$ by minimizing $P_\zeta(c = 1|s_t, z)$.

## C  SENSITIVITY OF OFFLINE DATA SIZE

To investigate the sensitivity of the offline data size, we conducted an experiment on Ant and Hopper environments using various proportions of the original offline demonstration data. Specifically, we tested ratios relative to the original size of the offline data, including 1.0, 0.5, 0.2, and 0.1. We report the metric of the ratio between Per-timestep Reward (PtR) and #Violations (PtR/#V ($\times 10^3$)) as detailed in our main paper, and present the results in Table 2.

It's evident that the PtR/#V decreases as we decrease the size of the offline data. Initially, there's a slight performance drop when half of the offline data is used, and this decline persists with further reductions in data size. This sensitivity is more notable in the ant environment, where there's a low total number of violations. The performance on the Ant environment drops by more than half when only 10% of the offline data is utilized. However, even with an extremely small offline data size, the performance on Hopper remains acceptable.

