# OpenReview forum: "Skill-based Safe Reinforcement Learning with Risk Planning"
_ICLR.cc/2025/Conference — Submitted to ICLR 2025_

### Official Review · Reviewer_JVhb · 2024-11-01

**Soundness:** 4
**Presentation:** 3
**Contribution:** 2
**Rating:** 3
**Confidence:** 5

**Summary:**

The paper considers safe reinforcement learning from demonstration problem where there is offline demonstration dataset available. The proposed method, SSkP, seeks to draw both skill encoder/policy and skill safety information from the offline data, and during online learning plan with the skill safety information to lower safety violations. The authors conduct experiments on four simulated MuJoCo environments and show SSkP can achieve higher performance in terms of performance-risk ratio, i.e., given the same amount of safety violation, SSkP’s performance outperforms. The paper is generally well-written and the experiments support the the paper’s claims, but I find important details about the method missing, and therefore wish authors to clarify.

**Strengths:**

1. The paper looks into an important topic of safety in RL learning process (not only the execution phase but also the training phase), and presents the novel algorithm, SSkP, with mild assumptions of offline data available (some data is labeled as unsafe while all other data is assumed unlabeled).

2. The writing is high-quality; therefore, it is easy for me to follow the whole paper’s logic. The algorithm pseudocodes were very clear about what the procedures are.

**Weaknesses:**

1. [biggest question] Based on Algorithm 1 and 2, it seems the risk planning is just choosing the safest skill to activate given the current state without considering how the skill contributes to the task completion. Line 4 of algorithm2 passes in the policy and the risk predictor, but algorithm 1 does not utilize the policy at all and rather samples from the skill prior q_\psi. In this case, wouldn’t the agent just choose safe but task-irrelevant skills? For example, to avoid colliding with anyone, an autonomous vehicle would just stop and stay.

2. From my understanding, the online RL only updates the skill policy network that chooses skills based on state and keeps the skill policy that chooses actions based on state and skill fixed (this only learns from the offline dataset). Why? Do you assume the action policies learned from offline dataset are very good? How much demonstration was used in the experiments and has the authors done scale experiments with different amount of offline data?

3. The paper seems to be lacking a “preliminary” section where the authors should show prior work that is directly utilized in the method, such as the PU learning and skill learning via skill-conditioned learning from demonstration. These are important prior knowledge to understand the whole method. More importantly, the paper currently mixes previous methods with its own, novel components, making it hard to judge the novel parts of the work. For example, it seems like the skill risk predictor is novel and all other skill learning from demonstration is using the previous work. However, the method to learn the skill risk predictor again is a prior method PU.

3.1. How is \lambda determined? It seems an essential hyperparameter for PU to work.

3.2. In Equation 6, why don’t the authors consider allowing covariance? Covariance is an important factor for CEM to accurately represent the correct shape of the distribution.

4. The experiment domains seem not diverse enough (all locomotion tasks) while there are many good safe RL benchmarks such as safety-gym and safety-gymnasium.

4.1. Line 415 states “SMBPO demonstrates a similar inability as CPQ in terms of learning a good policy to maximize the expected reward”. I believe this is not true as the SMBPO final performance is very close to SSkP (much higher than CPQ), and the Table 1’s result on Ant shows that too.

5. The related work is mostly comprehensive but lacks a clear description of how the current work distinguishes itself from prior work, in each of the paragraphs of the related work. I strongly recommend authors discuss this to show the uniqueness of the proposed approach.

5.1. On line 96, the authors stated “Reinforcement Learning from Demonstration, also known as Imitation Learning” – I believe this is generally not the well-accepted definition of imitation learning. Imitation Learning generally refers to methods like behavior cloning and inverse reinforcement learning. Reinforcement Learning from Demonstration is a series of work on its own.

5.2. There has been work on extracting safety information from demonstrations, such as [1]. The authors could discuss the differences between their work and prior work.

[1] Yang, Y., Chen, L., Zaidi, Z., van Waveren, S., Krishna, A., & Gombolay, M. (2024, March). Enhancing Safety in Learning from Demonstration Algorithms via Control Barrier Function Shielding. In Proceedings of the 2024 ACM/IEEE International Conference on Human-Robot Interaction (pp. 820-829).

**Questions:**

See weakness

---

> ### Author Response · Authors · 2024-12-01
>
> We sincerely thank the reviewer for dedicating their time and effort to thoroughly reviewing our work and providing detailed feedback.
>
> **Q1.** About the risk planning.
>
> **Answer:** First, we apologize for the **typo** in Algorithm 1, where the skills should be sampled from the **skill policy $\pi_\theta$** rather than the **skill prior $q_\psi$**. The skill policy $\pi_\theta$ is designed not only to identify a safe policy but also to maximize the return. Consequently, the risk planning aims not only to select safe skills but also to identify the safest skill within a close neighborhood of the skill that maximizes the return, thereby balancing the tradeoff between safety and performance.
>
> **Q2.** About the skill policy.
>
> **Answer:** This question may be raised from the same typo in Algorithm 1. We do not assume that the skill model learned from the offline demonstrations is good enough. Instead, we regularize the skill policy $\pi_\theta$ using the skill prior $q_\psi$. Beyond this, the skill policy is trained with refined skills obtained through risk planning and data collected from online interactions with the environment. For the offline demonstrations, we used $10^5$ timesteps of data collected from a fully trained safe RL agent. The impact of the demonstration data scale is examined in Section C of the appendix. We did not use a larger dataset to prevent abuse on offline demonstrations.
>
> **Q3.** About the contributions and related works.
>
> **Answer:** Due to page limitations, we provided only a brief discussion of prior works, focusing on minimal details to clearly distinguish previous works from our contributions. The relevant prior works are primarily discussed in the second paragraph of Section 4.1, and in the context of the principles of PU learning. All other components, including the novel application of PU classification for distinguishing safe and unsafe states, the risk predictor, the risk planning module, and the overall training framework, represent our original contributions. However, we would like to clarify our contributions in the related works section and provide a detailed discussion of the work mentioned by the reviewer [1].
>
> **Q4.** About the hyperparameter $\lambda$.
>
> **Answer:** We estimate the class prior $\lambda$ using the label frequency under the SCAR assumption. Specifically, $\lambda$ is calculated based on the proportion of safety violations in the offline demonstrations, as discussed in detail in [2, 3].
>
> **Q5.** About the covariance.
>
> **Answer:** We use a diagonal covariance matrix to improve computational efficiency and reduce sampling complexity. Additionally, previous studies have demonstrated the advantages of using a diagonal covariance matrix in similar contexts [4,5].
>
> **Q6.** About the environments.
>
> **Answer:** We conducted our experiments on MuJoCo locomotion tasks because they are increasingly popular and present a significant challenge for training safe RL agents, as any safety violation leads to immediate failure. In contrast, many other popular safe RL tasks, such as those in Safety-Gym, adopt a different setup. These tasks aim to train an optimal safe RL agent that can tolerate a limited number of cumulative safety violations within a predefined threshold.
>
> **Q7.** About the experimental results.
>
> **Answer:** Referring to Figure 3, SMBPO does not achieve comparable final results to our SSkP. The results in Table 1 demonstrate a tradeoff between cumulative episodic reward and cost. Specifically, the PtR/#V ratio for SMBPO is high in the Ant environment because SMBPO's episodic reward increases rapidly during the early training stages, while the total number of safety violations remains low. However, SMBPO ultimately does not reach the performance level achieved by SSkP. Therefore, we argue that SMBPO shares a similar limitation with CPQ, exhibiting low reward and low cost.
>
> **Q8.** About the terminology of imitation learning.
>
> **Answer:** The interchangeable use of the terms "Learning from Demonstration" (LfD) and "Imitation Learning" (IL) may date back to the original 1996 paper on Learning from Demonstration [6], as discussed in the last paragraph of the first page. Several subsequent works in this field have also used LfD and IL interchangeably [7,8]. This interchangeable usage is further mentioned in an MIT course material [9]. While this interpretation may not represent a universal definition adopted by all works, we believe it is appropriate to use the terms interchangeably in this context.

---

> > ### Comment · Reviewer_JVhb · 2024-12-01
> >
> > Thank you for providing answers to my questions and concerns! Quick question: it looks like the authors did not update the paper's writing in response to my Q1-Q3 and other clarification questions. Are the authors planning to do so? Even if it is not allowed to upload PDF anymorenow, talking about any plan to improve the presentation of the method will be very helpful for me and other reviewers to assess further.

---

> > > ### Author Response · Authors · 2024-12-01
> > >
> > > We appreciate the reviewer’s follow-up discussion.
> > >
> > > **Q9.** About the future plans for improving the paper."
> > >
> > > **Answer:** In response to the reviewer’s suggestions for improving the clarity of our paper, we will address the identified issues as follows: First, we will correct the typo where $q_\psi$ was incorrectly used instead of $\pi_\theta$, as this could lead to significant misunderstandings of our approach. In the Related Works section, we will incorporate the reviewer’s feedback by providing more details about the prior works referenced in our paper, and by offering a clear description of our contributions in comparison to these works. Additionally, we plan to include newly explored works on leveraging safety information from demonstrations to further enrich the related works. Regarding the interchangeable use of the terms LfD and IL, we plan to keep these terms but will include proper citations to clarify the source of this interpretation. We welcome any further suggestions from the reviewer to improve our paper.

---

> ### Author Response · Authors · 2024-12-01
>
> [1] Yang, Yue, et al. "Enhancing Safety in Learning from Demonstration Algorithms via Control Barrier Function Shielding." Proceedings of the 2024 ACM/IEEE International Conference on Human-Robot Interaction. 2024.
>
> [2] Elkan, Charles, and Keith Noto. "Learning classifiers from only positive and unlabeled data." Proceedings of the 14th ACM SIGKDD international conference on Knowledge discovery and data mining. 2008.
>
> [3] Bekker, Jessa, and Jesse Davis. "Learning from positive and unlabeled data: A survey." Machine Learning 109.4 (2020): 719-760.
>
> [4] Zhang, Zichen, et al. "A simple decentralized cross-entropy method." Advances in Neural Information Processing Systems 35 (2022): 36495-36506.
>
> [5] Ho, Jonathan, Ajay Jain, and Pieter Abbeel. "Denoising diffusion probabilistic models." Advances in neural information processing systems 33 (2020): 6840-6851.
>
> [6] Schaal, Stefan. "Learning from demonstration." Advances in neural information processing systems 9 (1996).
>
> [7] Pertsch, Karl, et al. “Demonstration-Guided Reinforcement Learning with Learned Skills.” 5th Annual Conference on Robot Learning, 2021
>
> [8] Liu, Minghuan, et al. “Plan Your Target and Learn Your Skills: State-Only Imitation Learning via Decoupled Policy Optimization.” 2021
>
> [9] Russ Tedrake. Underactuated Robotics: Algorithms for Walking, Running, Swimming, Flying, and Manipulation (Course Notes for MIT 6.832). https://underactuated.csail.mit.edu/

---

### Official Review · Reviewer_np9T · 2024-11-04

**Soundness:** 3
**Presentation:** 2
**Contribution:** 2
**Rating:** 6
**Confidence:** 3

**Summary:**

The paper introduces a novel approach called Safe Skill Planning (SSkP) for safe reinforcement learning (Safe RL). SSkP enhances safe RL by leveraging offline demonstration data through a two-stage process. The first stage involves learning a skill risk predictor using Positive-Unlabeled (PU) learning from offline demonstrations. The second stage uses this predictor to develop a risk planning process that enhances online safe RL, learning a risk-averse safe policy efficiently while adapting the skill risk predictor to the environment. Experiments in various robotic simulation environments show that SSkP outperforms other state-of-the-art safe RL methods.

**Strengths:**

- The paper proposes a new method, SSkP, that combines skill learning with risk planning for safe RL.
- The approach uses a two-stage process that first learns from offline data and then applies it to online environments, which is efficient and reduces potential damage to physical environments.
- The risk planning process is a simple yet effective method for generating safer skill decisions, enhancing safe exploration and learning.
- The method adapts the skill risk predictor to online environments in real-time, showing the ability to learn and adjust on the fly.

**Weaknesses:**

- The effectiveness of SSkP relies heavily on the quality and quantity of offline demonstration data, which may not always be available or reliable.
- The two-stage process and the integration of multiple components (skill model, risk predictor, risk planning) might make the approach more complex to implement and understand.
- The paper primarily focuses on robotic simulation environments, and it's unclear how well SSkP would generalize to other types of environments or real-world applications.
- The paper does not discuss the computational cost of the risk planning process, which could be high, especially with large state and action spaces.
-  As shown in the sensitivity analysis, the performance of SSkP decreases with smaller offline data sizes, indicating a potential weakness in scenarios with limited data.

**Questions:**

- How does SSkP handle partially incorrect or noisy offline demonstration data?
- What are the computational requirements of SSkP, and how does it scale with the size of the state and action spaces?
- How does SSkP compare to other safe RL methods in terms of computational efficiency?

---

> ### Author Response · Authors · 2024-12-01
>
> We deeply appreciate the reviewer’s positive feedback and thorough review of our work.
>
> **Q1.** About the quality of demonstration data.
>
> **Answer:** Our proposed SSkP framework does rely on the quality of demonstration data, as it uses expert demonstrations to learn safe skills. However, it still able to partially handle incorrect or noisy demonstration data. The risk predictor is initially pretrained on offline demonstration data but is further fine-tuned during the online stage by collecting additional Positive-Unlabeled (PU) data through interactions with the environment. Moreover, although the skill model is pretrained on potentially noisy demonstration data, the safe skill policy is learned during the online stage, which helps mitigate the negative impact of noisy data.
>
> **Q2.** About the computational requirements.
>
> **Answer:** Our experiments were conducted on 8 dedicated servers, each equipped with 4 Intel Core processors, 32 GB of RAM, and NVIDIA GeForce RTX 2080 GPUs with 11 GB of VRAM. While there are no strict computational requirements, sufficient RAM is needed to store the data, and an efficient GPU is recommended for training the skill model. Notably, the computational cost does not increase significantly as the action and state spaces grow larger.
>
> **Q3.** About the computational efficiency.
>
> **Answer:** Our proposed SSkP framework does not demonstrate a significant advantage in computational efficiency compared to other safe RL methods. The primary computational cost arises from the additional training of the skill model, which is not required in conventional safe RL methods, and the online safe policy training, which is common to most safe RL approaches. The risk planning component does not substantially increase the overall computational cost. Importantly, the skill model facilitates the training of the safe RL agent by effectively balancing performance and safety. Additionally, the SSkP method reduces safety violations during training, thereby minimizing potential harm to the environment.

---

### Official Review · Reviewer_pb5r · 2024-11-04

**Soundness:** 2
**Presentation:** 2
**Contribution:** 1
**Rating:** 3
**Confidence:** 3

**Summary:**

The paper proposes a method that leverages additional offline data to train skills and a risk predictor and refines the risk predictor while collecting online data, ultimately training a skill-based policy.
For safe exploration, the proposed method uses a risk planning strategy, similar to a cross-entropy method (CEM).

**Strengths:**

- The paper introduces the idea of using offline data for training skills and a risk predictor.
- The paper is well-organized and easy to read.

**Weaknesses:**

# Major Weaknesses
- The contributions are relatively limited.
    - Each proposed module (skills [1,2], risk predictors [3, 4, 5]) is from existing methods.
        - If there were any novel training techniques, the author should have highlighted them, but it seems there is nothing new.
    - The motivation for the proposed method is ambiguous, and in particular, it is unclear which parts of the proposed method improve upon existing methods (lines 44–48).
    - Also, there is no theoretical analysis to motivate it.
- The experimental validation is weak, with only four tasks and using just three seeds per task, which is insufficient for robust evaluation.
    - Additional experiments on safe RL benchmarks, such as Safety Gymnasium [6], are necessary for a comprehensive evaluation.
    - The baseline methods, primarily from 2021 to 2022, appear somewhat outdated, so it would be nice to include a state-of-the-art method.
- Ablation studies are not sufficient to show the roles of each module.
    - Although the current ablation study has focused on risk planning, it seems obvious that eliminating risk planning would result in unsafe outcomes, as it does not take into account costs when training the skill-based policy.
    - The training of the risk predictor and the learning of skills appear to be significantly influenced by the proportion of cost violations or the diversity of demonstrations. Therefore, a performance comparison based on different demonstrations seems essential.
    - Additionally, since using skills represents a distinct contribution, comparative experiments between employing a skill-based policy and using a direct-action policy are also necessary.

# Minor Weaknesses
- In line 178, it is essential to clarify that $\zeta$ in $P_\zeta(C|s_t,z_t)$ represents the neural network parameters to avoid confusion.
- For reproducibility, Section 4.1 should include at least loss functions or the algorithms used for training the encoder and decoder of the skills.

# References
- [1] Campos, Víctor, et al. "Explore, discover and learn: Unsupervised discovery of state-covering skills." International Conference on Machine Learning. PMLR, 2020.
- [2] Eysenbach, Benjamin, et al. "Diversity is all you need: Learning skills without a reward function." arXiv preprint arXiv:1802.06070 (2018).
- [3] Bharadhwaj, Homanga, et al. "Conservative safety critics for exploration." arXiv preprint arXiv:2010.14497 (2020).
- [4] Srinivasan, Krishnan, et al. "Learning to be safe: Deep RL with a safety critic." arXiv preprint arXiv:2010.14603 (2020).
- [5] Thananjeyan, Brijen, et al. "Recovery RL: Safe reinforcement learning with learned recovery zones." IEEE Robotics and Automation Letters 6.3 (2021): 4915-4922.
- [6] Ji, Jiaming, et al. "Safety gymnasium: A unified safe reinforcement learning benchmark." Advances in Neural Information Processing Systems 36 (2023).

**Questions:**

1. Given the inclusion of a planning module, this paper appears to suggest that execution time may increase as a potential drawback. Are there any results measuring execution times based on different planning parameters?

2. Is there a part of the method that improves skills online? If not, it seems that the initial skill sets the upper bound for performance.

3. When training a skill-based policy, there appears to be no component accounting for safety, which might result in a high likelihood of choosing actions far from safety during risk planning. How has this issue been addressed? According to lines 334–336, penalties are applied to encourage actions that are closer to a prior. Does this approach effectively resolve the concern?

---

> ### Author Response · Authors · 2024-12-01
>
> We sincerely appreciate the reviewer’s time, effort, and thoughtful review of our work.
>
> **Q1.** About the efficiency of planning.
>
> **Answer:** We do not claim that incorporating the planning model will increase execution time. Instead, by leveraging skill learning, we aim to facilitate the learning of the safe policy through the risk planning process. Since the planning is performed offline and primarily involves interactions with the risk predictor, it does not significantly increase the overall training time per iteration. The primary time consumption comes from training the skill model on offline demonstration data, and the online training of the safe policy through interactions with the environment.
>
> **Q2.** About the improvement of skills.
>
> **Answer:** Although the risk planning process itself is conducted offline, the refined skills generated through the risk planning has online interactions with the environment. Additionally, the risk predictor is further refined using new data collected during these online interactions, leading to continuous improvement of the learned skill model.
>
> **Q3.** About the safety of the policy.
>
> **Answer:** As the reviewer mentioned, the safe policy $\pi_\theta$ is regularized with the skill prior $q_\psi$ using a KL-divergence term, which contributes to the safety of the policy and represents one of the methods to ensure its safety. Additionally, the risk planning module identifies the safest skill within the close neighborhood of the skills generated by the safe policy, aiming to balance the tradeoff between performance and safety.
>
> **Q4.** About the contribution.
>
> **Answer:** While there is extensive prior work on skills in RL, applications on skills learning in safe RL remains limited, and no prior work has explored designing a planning process to enhance learned skills in safe RL. Moreover, although risk prediction is widely used in safe RL, it is innovative to frame the safe RL problem as a Positive-Unlabeled (PU) learning problem, which distinguishes conventionally treated safe states from potentially unsafe unlabeled data.
>
> **Q5.** About the ablation study.
>
> **Answer:** We have accounted for the training of the skill-based policy in our approach. In the ablation variants that **exclude risk planning**, only the risk planning component is removed, while all other components remain unchanged. Consequently, the skills are not refined through risk planning but are instead directly sampled from the safe skill policy. The impact of **demonstration data quality** is discussed in Section C of the appendix. As observed, the performance—particularly the quality of the pretrained skills and the risk predictor—decreases as the size of the demonstration data decreases. Regarding the ablation variant **without skill learning**, since skill learning serves as the base method of our imitation learning approach, it is essential to derive a safe behavior policy from offline demonstrations in the form of skills. Replacing the skill-based policy with a direct-action policy outside the safe skill learning framework is extremely difficult.

---

> > ### Comment · Reviewer_pb5r · 2024-12-03
> > **Response by Reviewer pb5r**
> >
> > Thanks for the response. Due to the following reasons, I will stick with the initial score:
> >
> > 1. I don't understand why the authors said that the risk planning is performed offline.
> > Algorithm 2 of the paper states that the risk planning is performed at every step in online learning process.
> >
> > 2. I think the KL regularization term is critical to the safety.
> > Still, the authors need to check which part improves the safety performance.
> >
> > 3. Authors mentioned that the risk planning process refines the skill. However, the risk planning does not refine the skill, but select a proper skill. This is not a proper answer to my question and is a common technique in the field of robotic control.
> >
> > 4. PU learning problem is quite heuristic, but ablation studies on this method do not performed widely.
> > The authors performed ablation on a proportion of PU data, but the results should be compared with the quality of PU data.
> >
> > 5. Also, the experiment parts are not improved.

---

> > > ### Author Response · Authors · 2024-12-03
> > >
> > > We appreciate the reviewer’s follow-up discussion.
> > >
> > > **Q6.** About clarification on risk planning
> > >
> > > **Answer:** We apologize for any confusion. By stating that risk planning is offline, we mean that the process of improving skills in risk planning does not involve interactions with the environment; all skills are sampled offline. However, the output skill from the risk planning stage interacts with the environment online by being decoded into an action sequence and executed within the environment.
> > >
> > > **Q7.** About skills from risk planning
> > >
> > > **Answer:** We agree that planning is a common technique in the literature. However, its functionality extends beyond simply selecting an appropriate skill. Otherwise, we could merely choose the skill with the lowest risk prediction, as demonstrated by the ablation variant with naive planning, which exhibits degraded performance. First, in risk planning, we iteratively optimize the skill distribution by reconstructing it to achieve lower risk predictions. Second, these optimized skills are then utilized in online interactions with the environment, enabling further training of a safe skill policy.
> > >
> > > **Q8.** About additional ablation studies
> > >
> > > **Answer:** We will consider dropping the KL regularization term as an ablation variant to illustrate the contribution of risk planning to safety. However, we believe it is unnecessary to examine the quality of PU data, as all data were collected using either a safe RL agent or the current running policy. Modifying the demonstration data would compromise the foundational assumptions of our method.

---

### Official Review · Reviewer_agFy · 2024-11-04

**Soundness:** 2
**Presentation:** 2
**Contribution:** 2
**Rating:** 3
**Confidence:** 3

**Summary:**

The paper presents a novel approach using learned skills to improve online exploration in CMDPs given available offline data. In the offline phase, the presented method leverages the algorithm presented in [1] to learn skills as high-level actions. To enable safe learning, the authors employ a safety classifier $P_\xi(c|s_t,z_t)$ that takes as input $s_t$ and the skill $z_t$ using a similar PU learning approach similar to [2]. In the online exploration phase, the authors use SAC to learn a skills policy $\pi_\theta(z_t | s_t)$ to maximize rewards. In each environment step, the learned skills policy $\pi_\theta(z_t | s_t)$ is used to initialize the plan for a risk planner based on CEM. The planner optimizes for safe skills by minimizing the safety classifier cost. In the experimental section, the paper is compared to recovery RL[3], CPQ[4] and SMBPO[5].

The contributions of the paper can be summarized in the following points
- Exploring the skill learning method from [1] in a safe RL setting.
- Learning safety classifier using PU sampling.
- Introducing a risk planner to optimize safe skills by minimizing safety classifier costs.

[1] Pertsch, Karl, Youngwoon Lee, and Joseph Lim. "Accelerating reinforcement learning with learned skill priors." Conference on robot learning. PMLR, 2021.

[2] Xu, Danfei, and Misha Denil. "Positive-unlabeled reward learning." Conference on Robot Learning. PMLR, 2021.

[3] Thananjeyan, Brijen, et al. "Recovery rl: Safe reinforcement learning with learned recovery zones." IEEE Robotics and Automation Letters 6.3 (2021): 4915-4922.

[4] Xu, Haoran, Xianyuan Zhan, and Xiangyu Zhu. "Constraints penalized q-learning for safe offline reinforcement learning." Proceedings of the AAAI Conference on Artificial Intelligence. Vol. 36. No. 8. 2022.

[5] Thomas, Garrett, Yuping Luo, and Tengyu Ma. "Safe reinforcement learning by imagining the near future." Advances in Neural Information Processing Systems 34 (2021): 13859-13869.

**Strengths:**

The paper investigates an interesting setting of using offline data to enable RL for online exploration in constrained settings. This topic is crucial for enabling RL to explore in constrained real-world scenarios. In this setting, the paper explores using skills in safe RL, which is a relatively unexplored combination. The paper provides a clear description of the algorithm, which aids in understanding the advantages and disadvantages of the proposed method.

**Weaknesses:**

- The impact of using skills needs to be ablated. It is unclear whether using skills in this context is useful. Is it beneficial to only act on each H step in a constrained setting? The authors should ablate using the presented planning and policy learning combination on low-level actions.

- The motivation behind using PU learning to learn the safety classifier is unclear. The authors use the classifier as a cost-to-go function, which would typically be learned as a safety critic using approximate dynamic programming [1,2]. It's unclear whether PU learning can be better than approximate dynamic programming in capturing the temporal aspect of the cost.

- The paper claims to learn a safe policy when the policy is not optimized to learn safe behavior but to maximize rewards. This is clear in the ablation studies where the policy has a much worse behavior regarding adherence to constraints without the risk planner. However, the paper mentions learning a safe policy on multiple occasions, which needs to be clarified.

- Unfair comparison with baselines. SMBPO should also be pretrained to ensure a fair comparison. Even training it from scratch, SMBPO has very similar performance in 2 environments. Additionally, the recovery RL baseline is not tuned to the benchmark used. However, recovery RL is sensitive to the $\epsilon$ parameter that triggers the recovery policy [1]. Even with an unfair comparison, recovery RL has a similar performance to the presented method in 2 environments. Also, recovery RL only uses the recovery policy (its version of the risk planner) when a trigger condition is fulfilled, unlike the presented method, where the risk planner is used in every step. It's thus hard to judge whether the use of skills is the real reason for the difference in performance or the use of planning in every environment step.

- The absence of an explanation on how the offline data was collected is a significant gap. Details such as the number of samples, the ratio of samples with constraint violations, and other relevant information are crucial for the sake of reproducibility and should be provided.

[1] Thananjeyan, Brijen, et al. "Recovery rl: Safe reinforcement learning with learned recovery zones." IEEE Robotics and Automation Letters 6.3 (2021): 4915-4922.

[2] Srinivasan, Krishnan, et al. "Learning to be safe: Deep rl with a safety critic." arXiv preprint arXiv:2010.14603 (2020).

**Questions:**

- How would the method perform if we replaced skills with low-level actions? It's unclear whether the performance is due to using skills or a CEM planner to minimize constraint cost in every environment step.
- Why not use a safety critic similar to [1,2] instead of using PU learning for the safety classifier?
- In lines (19, 128, 307), you mention learning a safe policy $\pi_\theta$; however, as far as I understand, the policy only optimizes the policy to maximize rewards. Can you explain what makes the policy safe?
- Can you explain equation 7?
- Can you provide more details on offline data, including how it was collected and the percentages of constraint-violating samples in the data?

Additional remarks:

- The abbreviation PU in used in the abstract without prior introduction
- LfD is an abbreviation for Learning from Demonstrations, not Reinforcement Learning from Demonstrations (line 42)
- In line 218 in algorithm1, you mention you sample ($\mu_0, \Sigma_0$) from $q_\psi$, which contradicts the second paragraph in 4.2.1.
- in line 9 in algorithm, what exactly is pair $\mathcal{P}$?
- In the conclusion, you mention learning safe behavior patterns from demonstrations. However, the skill model is not optimized to teach safe skills, just skills as high-level representations of actions. I think this can be confusing.

[1] Thananjeyan, Brijen, et al. "Recovery rl: Safe reinforcement learning with learned recovery zones." IEEE Robotics and Automation Letters 6.3 (2021): 4915-4922.

[2] Srinivasan, Krishnan, et al. "Learning to be safe: Deep rl with a safety critic." arXiv preprint arXiv:2010.14603 (2020).

---

> ### Author Response · Authors · 2024-12-01
>
> We are truly grateful to the reviewer for their time, effort, and detailed consideration in reviewing our work.
>
> **Q1.** About the significance of skills.
>
> **Answer:** We did not include an ablation variant **without** skill learning because skill learning serves as the base method for our imitation learning approach. In order to learn a safe behavior policy from offline demonstrations, we must first learn safe skills from the offline demonstration data. Without this skill learning framework, it would be extremely challenging to replace the learned skills with low-level actions for executing safe RL tasks. The potential performance benefits of CEM are analyzed in the ablation study presented in Section 5.3.
>
> **Q2.** About the motivation of using PU learning.
>
> **Answer:** The reason we use Positive-Unlabeled (PU) learning instead of a safety critic is discussed in Section 4.1.1. Traditional safe RL methods [1,2] approach the learning of safety constraints as Positive-Negative (PN) learning, where states without safety violations are **strictly** treated as **safe**. However, leveraging the properties of safe RL environments, we argue that even states **without** immediate safety violations can lead to future violations. For instance, an autonomous vehicle approaching an obstacle at high speed may not yet have collided but poses a significant risk of a safety violation. To address this, we employ PU learning to train a risk predictor, treating states without safety violations as **unlabeled** data, rather than learning a safety critic.
>
> **Q3.** About the function of safe policy.
>
> **Answer:** The safe policy $\pi_\theta$ ensures safety through two ways. First, instead of directly outputting actions, it samples skills based on the current state $s_t$. As outlined in the final paragraph of Section 4.2.2, the safe policy is regularized with a KL divergence term relative to the skill prior $q_\psi$, promoting safety through this regularization. Second, while $\pi_\theta$ is optimized to maximize rewards, the skills sampled from the policy are subjected to a risk planning procedure that refines them into safer skills within a close neighborhood, balancing the tradeoff between performance and safety.
>
> **Q4.** About the decision pairs $\mathcal{P}$.
>
> **Answer:** The $\mathcal{P}$ in line 9 of the algorithm represents the decision pairs described in Eq. (7). In Eq. (7), these decision pairs $\mathcal{P}$ consist of state-skill pairs $(s_t, z_t)$ over a horizon of length $H$ through interacting with the environment. The skill $z_t$ in the initial state-skill pair $(s_t, z_t)$ is refined using the risk planning procedure, while subsequent skills are sampled from the skill prior $q_\psi(\cdot|s_t)$ because only the first skill is processed using risk planning. The decision pairs $\mathcal{P}$ are then stored in the PU data to further refine the risk predictor.
>
> **Q5.** About the collection of offline demonstration.
>
> **Answer:** We collected the offline demonstration data using a fully trained RL agent. Specifically, a fully trained SMBPO [3] agent, trained for $10^7$ timesteps, was utilized to generate $10^5$ timesteps of data for the offline demonstrations. Among the collected trajectories, approximately 10% contain safety violations.
>
> **Q6.** About additional marks.
>
> **Answer:** The skill model effectively learns safe behavior from the offline demonstrations. Additionally, the safe skills are further refined through risk planning with the help of the risk predictor, ensuring the safety of the RL agent while functioning beyond only the high-level representation of actions. Regarding the other typos noted by the reviewer, including those related to PU, LfD, and $q_\psi$ in Algorithm 1, we appreciate the reviewer’s attention to detail and will address these issues in the revised version.
>
> [1] Thananjeyan, Brijen, et al. "Recovery rl: Safe reinforcement learning with learned recovery zones." IEEE Robotics and Automation Letters 6.3 (2021): 4915-4922.
>
> [2] Srinivasan, Krishnan, et al. "Learning to be safe: Deep rl with a safety critic." arXiv preprint arXiv:2010.14603 (2020).
>
> [3] Thomas, Garrett, Yuping Luo, and Tengyu Ma. "Safe reinforcement learning by imagining the near future." Advances in Neural Information Processing Systems 34 (2021): 13859-13869.

---

> > ### Comment · Reviewer_agFy · 2024-12-03
> >
> > Thanks for the reply. I'm sticking with my initial score due to the following reasons.
> >
> > - There is no ablation to measure the effect of using high-level skills. The ablation study referred to by the authors only studies the impact of risk planning. However, when comparing the results with recovery RL, which is the primary baseline, the CEM planner in recovery RL  is triggered only under certain conditions, while in this paper, it is triggered at every environment step. So, it is difficult to disentangle the effect of each module in the end results.
> > - In safety critics, approximate dynamic programming is usually used to use the safety critic as a bootstrap. The safety critic $Q^{\pi_\theta}(s,a)$ captures the likelihood of the policy $\pi_\theta$ violating constraint starting state $s$, which would be high in risky states even if the constraints are not violated in $s$ [1]. Thus, the utility of PU learning here is not very clear, and it has also not been ablated.
> > - In my understanding, $\pi_\theta$ is not trained to minimize constraint violation but to maximize rewards. Adherence to constraints is done by the risk planner. As such, it is still unclear why $\pi_\theta$ is referred to as a safe policy.
> > - The authors have not addressed the fairness of their comparisons to the baselines.
> >
> > [1] Thananjeyan, Brijen, et al. "Recovery rl: Safe reinforcement learning with learned recovery zones." IEEE Robotics and Automation Letters 6.3 (2021): 4915-4922.

---

### Meta-Review · Area_Chair_J2Wu · 2024-12-20

**Metareview:**

While the paper addresses an interesting and important problem of using offline data for safe RL exploration with skills, the novelty and empirical validation are insufficient for acceptance. Key issues include the lack of ablation on the impact of using skills versus low-level actions, unclear motivation for the choice of learning over standard dynamic programming for the safety classifier, misleading claims about learning a "safe policy," unfair comparisons with baselines, and a lack of crucial details regarding the offline dataset. The performance gains are also not convincing enough to overcome these weaknesses, especially given the unfair baseline comparisons. Therefore, despite the potential of the problem setting, the current submission is not ready for publication and requires substantial improvements to address these concerns.

**Additional Comments On Reviewer Discussion:**

While some concerns from the reviewers are addressed, the paper is benefitted by another revision before acceptance.

---

### Decision · Program_Chairs · 2025-01-22

Reject